# Comprehensive Analysis of Tumour Sub-Volumes for Radiomic Risk Modelling in Locally Advanced HNSCC

**DOI:** 10.3390/cancers12103047

**Published:** 2020-10-19

**Authors:** Stefan Leger, Alex Zwanenburg, Karoline Leger, Fabian Lohaus, Annett Linge, Andreas Schreiber, Goda Kalinauskaite, Inge Tinhofer, Nika Guberina, Maja Guberina, Panagiotis Balermpas, Jens von der Grün, Ute Ganswindt, Claus Belka, Jan C. Peeken, Stephanie E. Combs, Simon Boeke, Daniel Zips, Christian Richter, Mechthild Krause, Michael Baumann, Esther G.C. Troost, Steffen Löck

**Affiliations:** 1OncoRay—National Center for Radiation Research in Oncology, Faculty of Medicine and University Hospital Carl Gustav Carus, Technische Universität Dresden, Helmholtz-Zentrum Dresden—Rossendorf, 01307 Dresden, Germany; Alex.Zwanenburg@nct-dresden.de (A.Z.); Karoline.Leger@uniklinikum-dresden.de (K.L.); Fabian.Lohaus@uniklinikum-dresden.de (F.L.); Annett.Linge@uniklinikum-dresden.de (A.L.); christian.richter@oncoray.de (C.R.); Mechthild.Krause@uniklinikum-dresden.de (M.K.); michael.baumann@dkfz-heidelberg.de (M.B.); esther.troost@uniklinikum-dresden.de (E.G.C.T.); steffen.loeck@oncoray.de (S.L.); 2German Cancer Research Center (DKFZ), Heidelberg and German Cancer Consortium (DKTK) Partner Site, 01307 Dresden, Germany; 3National Center for Tumor Diseases (NCT), Partner Site Dresden of the German Cancer Research Center (DKFZ), Faculty of Medicine and University Hospital Carl Gustav Carus and Technische Universität Dresden, 01307 Dresden, Germany; 4Department of Radiotherapy and Radiation Oncology, Faculty of Medicine and University Hospital Carl Gustav Carus, Technische Universität Dresden, 01307 Dresden, Germany; 5Department of Radiotherapy, Hospital Dresden-Friedrichstadt, 01067 Dresden, Germany; schreiber-an@khdf.de; 6German Cancer Research Center (DKFZ), Heidelberg and German Cancer Consortium (DKTK) Partner Site, 10117 Berlin, Germany; goda.kalinauskaite@charite.de (G.K.); ingeborg.tinhofer@charite.de (I.T.); 7Department of Radiooncology and Radiotherapy, Charité University Hospital, 10117 Berlin, Germany; 8German Cancer Research Center (DKFZ), Heidelberg and German Cancer Consortium (DKTK) Partner Site, 45147 Essen, Germany; nika.guberina@uk-essen.de (N.G.); maja.guberina@uk-essen.de (M.G.); 9Department of Radiotherapy, University Hospital Essen, Medical Faculty, University of Duisburg-Essen, 45147 Essen, Germany; 10German Cancer Research Center (DKFZ), Heidelberg and German Cancer Consortium (DKTK) Partner Site, 60596 Frankfurt, Germany; Panagiotis.Balermpas@usz.ch (P.B.); Jens.VonderGruen@kgu.de (J.v.d.G.); 11Department of Radiotherapy and Oncology, Goethe-University Frankfurt, 60596 Frankfurt, Germany; 12German Cancer Research Center (DKFZ), Heidelberg and German Cancer Consortium (DKTK) Partner Site, 81377 Munich, Germany; ute.ganswindt@i-med.ac.at (U.G.); Claus.Belka@med.uni-muenchen.de (C.B.); jan.peeken@tum.de (J.C.P.); Stephanie.Combs@tum.de (S.E.C.); 13Department of Radiation Oncology, Ludwig-Maximilians-Universität, 81377 Munich, Germany; 14Clinical Cooperation Group, Personalized Radiotherapy in Head and Neck Cancer, Helmholtz Zentrum, 81377 Munich, Germany; 15Department of Radiation Oncology, Medical University of Innsbruck, Anichstraße 35, A-6020 Innsbruck, Austria; 16Department of Radiation Oncology, Technische Universität München, 81675 Munich, Germany; 17Institute of Radiation Medicine (IRM), Helmholtz Zentrum München, 85764 Neuherberg, Germany; 18German Cancer Research Center (DKFZ), Heidelberg and German Cancer Consortium (DKTK) Partner Site, 72076 Tübingen, Germany; simon.boeke@med.uni-tuebingen.de (S.B.); Daniel.Zips@med.uni-tuebingen.de (D.Z.); 19Department of Radiation Oncology, Faculty of Medicine and University Hospital Tübingen, Eberhard Karls Universität Tübingen, 72076 Tübingen, Germany; 20Institute of Radiooncology—OncoRay, Helmholtz-Zentrum Dresden—Rossendorf, 01328 Dresden, Germany; 21German Cancer Research Center (DKFZ), 69120 Heidelberg, Germany

**Keywords:** radiomic, image-based risk modelling, machine learning, personalised therapy, radiation oncology

## Abstract

**Simple Summary:**

Radiomic risk models are usually based on imaging features, which are extracted from the entire gross tumour volume (GTV${}_{\mathrm{entire}}$). This approach does not explicitly consider the complex biological structure of the tumours. Therefore, in this retrospective study, we investigated the prognostic value of radiomic analyses based on different tumour sub-volumes using computed tomography imaging of patients with locally advanced head and neck squamous cell carcinoma who were treated with primary radio-chemotherapy. The GTV${}_{\mathrm{entire}}$ was cropped by different margins to define the rim and corresponding core sub-volumes of the tumour. Furthermore, the best performing tumour rim sub-volume was extended into surrounding tissue with different margins. As a result, the models based on the 5 mm tumour rim and on the 3 mm extended rim sub-volume showed an improved performance compared to models based on the corresponding tumour core. This indicates that the consideration of tumour sub-volumes may help to improve radiomic risk models.

**Abstract:**

Imaging features for radiomic analyses are commonly calculated from the entire gross tumour volume (GTV${}_{\mathrm{entire}}$). However, tumours are biologically complex and the consideration of different tumour regions in radiomic models may lead to an improved outcome prediction. Therefore, we investigated the prognostic value of radiomic analyses based on different tumour sub-volumes using computed tomography imaging of patients with locally advanced head and neck squamous cell carcinoma. The GTV${}_{\mathrm{entire}}$ was cropped by different margins to define the rim and the corresponding core sub-volumes of the tumour. Subsequently, the best performing tumour rim sub-volume was extended into surrounding tissue with different margins. Radiomic risk models were developed and validated using a retrospective cohort consisting of 291 patients in one of the six Partner Sites of the German Cancer Consortium Radiation Oncology Group treated between 2005 and 2013. The validation concordance index (C-index) averaged over all applied learning algorithms and feature selection methods using the GTV${}_{\mathrm{entire}}$ achieved a moderate prognostic performance for loco-regional tumour control (C-index: 0.61 ± 0.04 (mean ± std)). The models based on the 5 mm tumour rim and on the 3 mm extended rim sub-volume showed higher median performances (C-index: 0.65 ± 0.02 and 0.64 ± 0.05, respectively), while models based on the corresponding tumour core volumes performed less (C-index: 0.59 ± 0.01). The difference in C-index between the 5 mm tumour rim and the corresponding core volume showed a statistical trend (*p* = 0.10). After additional prospective validation, the consideration of tumour sub-volumes may be a promising way to improve prognostic radiomic risk models.

## 1. Introduction

The individualisation of radiation oncology is a major objective in modern cancer therapy [1]. Radiomics aims to characterise the tumour phenotype using advanced image features to predict patient-specific outcome. Commonly, radiomic features are computed and extracted using the entire gross tumour volume (GTV${}_{\mathrm{entire}}$) [2,3,4,5]. Such an approach assumes that the individual tumour is either homogeneous or heterogeneous, but uniformly distributed over the entire tumour volume. However, tumours are biologically complex and exhibit substantial spatial variation, e.g., in gene expression and in microscopic structure [6]. Such spatial variation may be caused by, e.g., hypoxia and necrosis which may appear in the tumour core and high cell proliferation and infiltrating tumour cell growth, which may occur at the tumour periphery [7]. Some of these regional tumour variations are apparent in imaging data, e.g., necrosis or tumour vascularisation detected by magnetic resonance imaging (MRI) or tumour hypoxia measured by ${}^{18}$F-fluoromisonidazole positron emission tomography (FMISO-PET) [8,9,10,11,12]. Furthermore, different regions within an individual tumour may differ in radio-sensitivity, which may depend on the distribution of cancer stem cells and localised genetic or molecular alterations [13,14]. In the case of head and neck squamous cell carcinoma (HNSCC), several studies have shown that the tumour micro-environment plays a major role for cancer development and progression [15]. Alsahafi et al. [16] showed that the poor response to therapy and the aggressive nature of HNSCC are not only caused by the complex alterations in intracellular signalling pathways, but are also influenced by the behaviour of the extracellular micro-environment. As consequences, such spatial variations may affect the performance of image-based risk models.

Thus far, only few studies have investigated and analysed specific tumour sub-volumes for radiomic risk modelling. Recently, Algohary et al. [17] showed that the combination of peri-tumoural and intra-tumoural radiomic features derived from prostate bi-parametric MR images leads to an improved risk assessment of prostate cancer patients. Furthermore, Grove et al. [18] showed that the expressions of 2-dimensional radiomic features computed on the rim of the tumour differed from those calculated on the tumour core. The ratio of tumour rim and core features led to an improved prediction of overall survival (OS) in non-small cell lung cancer patients. Wu et al. [6] identified clinically relevant tumour sub-volumes to characterise the regional heterogeneity of tumours in breast cancer patients based on dynamic contrast enhanced magnetic resonance imaging. The resulting risk models based on the identified sub-volumes also showed an improved outcome prediction compared to models based on the GTV${}_{\mathrm{entire}}$. In a further study, Wu et al. [19] identified different tumour sub-volumes using computed tomography (CT) and ${}^{18}$F-fluorodeoxyglucose PET (FDG-PET) imaging of lung cancer patients. It was shown that spatially distinct sub-volumes are linked to higher risk of recurrence compared to the GTV${}_{\mathrm{entire}}$, resulting in an improved model prediction of OS.

Aside from these initial findings, in most of the previously described studies, only individual clinical parameters or radiomic features (e.g., tumour volume) were investigated. Therefore, systematic investigations of the potential of radiomic risk models based on different tumour sub-volumes are still sparse.

In the present study, we systematically compared radiomic models based on two different sub-volumes of the GTV${}_{\mathrm{entire}}$, the outer tumour rim and the complementary tumour core, using pre-treatment CT imaging [20,21]. A multi-centre cohort of 291 patients with locally advanced HNSCC treated by primary radio-chemotherapy was considered. For the prediction of loco-regional tumour control (LRC), risk models were developed and independently validated. Patients were stratified into groups at low and high risk of loco-regional recurrence. Furthermore, we investigated the prognostic performance of the developed models for sub-groups of small and large tumours and extended the tumour rim beyond the GTV${}_{\mathrm{entire}}$ to account for potential sub-microscopic spread, leading to the clinical target volume [22].

## 2. Materials and Methods

### 2.1. Characteristics of Patient Cohorts

A retrospective multi-centre cohort consisting of 291 patients with histologically confirmed loco-regionally advanced HNSCC was used. All patients received primary radio-chemotherapy (RCT) and underwent a CT scan with or without contrast-enhancement for treatment-planning purpose. The multi-centre cohort was divided into an exploratory and a validation cohort by an approximate ratio of 2:1. In the exploratory cohort, 149 of the 206 patients were treated in one of the six Partner Sites of the German Cancer Consortium Radiation Oncology Group (DKTK-ROG) between 2005 and 2011 [23]. The remaining 57 patients were treated at the University Hospital Dresden (UKD) between 1999 and 2006. The validation cohort consisted of 85 patients from which 51 patients received their treatment within a prospective clinical trial (ClinicalTrials.gov Identifier: NCT00180180) at the UKD between 2006 and 2012 [9,12]. The remaining 34 patients were treated at the UKD or the Radiotherapy Centre Dresden-Friedrichstadt between 2005 and 2009 as well as at the University Hospital Tübingen between 2008 and 2013. Patient characteristics for the exploratory and the independent validation cohort are summarised in Table 1.

Radiomic risk models were developed to predict the primary clinical endpoint LRC, which was defined as the time from the first day of RCT to the date of loco-regional recurrence (event) or to the end of follow-up (censoring). Ethical approval for the multi-centre retrospective analyses of clinical and imaging data was obtained from the Ethics Committee at the Technische Universität Dresden (EK177042017, May 2017). All analyses were carried out in accordance with the relevant guidelines and regulations. Informed consent was obtained from all patients.

### 2.2. Tumour Sub-Volume Definition and Feature Computation

The analysis was divided into two subsequent steps, which are shown in Figure 1. The GTV${}_{\mathrm{entire}}$, i.e., the primary gross tumour volume, was manually delineated on each planning CT scan by a radiation oncologist using the CT image information only. Subsequently, the image voxel spacing was resampled using cubic spline image interpolation to an isotropic voxel size of 1.0 × 1.0 × 1.0 mm${}^{3}$ to correct for differences in voxel spacing and slice thickness between the cohorts [2,24].

Based on the delineated GTV${}_{\mathrm{entire}}$, two distinct sub-volumes were generated. The outer contour of the GTV${}_{\mathrm{entire}}$ was cropped by different margins (3 and 5 mm) to define the rim of the tumour (GTV${}_{3mm-rim}$ and GTV${}_{5mm-rim}$, respectively). The corresponding remaining sub-volumes were defined as tumour core (GTV${}_{3mm-core}$ and GTV${}_{5mm-core}$, respectively). The minimum core volume was restricted to 40% of the entire tumour volume to avoid disappearance of the core sub-volume in small tumours. Furthermore, the best performing tumour rim sub-volume was extended (GTV${}_{rim+ext}$) into surrounding tissue with different distances (1, 2, 3 and 5 mm) to assess the prognostic performance of the microscopic tumour extension.

Nine additional images were created by applying spatial filtering to the base image to emphasise image characteristics, such as edges and blobs. Eight additional images were created by applying a stationary coiflet-1 wavelet high-/low-pass filter along each of the three spatial dimensions. One further image was created by applying a Laplacian of Gaussian (LoG) filter consisting of five different filter kernel widths (1.0, 2.0, 3.0, 5.0 and 6.0 mm). Subsequently, the tumour mask was re-segmented to include only soft tissue voxels between −150 and 180 Hounsfield units, thereby removing voxels containing air or bone, which may affect feature expression. Features were implemented in compliance with the Image Biomarker Standardisation Initiative [25]. A total of 1538 features were computed and extracted from each sub-volume. A total of 18 statistical, 38 histogram-based and 95 texture features were calculated on the base image and on the nine transformed images. Moreover, 28 morphological features were determined on the base image only. The configuration settings for the image feature computation and extraction is summarised in Table A2.

### 2.3. Radiomic Risk Modelling

Radiomic risk models were developed using an end-to-end modelling framework, which consists of five steps: (I) feature pre-processing; (II) feature selection; (III) hyper-parameter optimisation; (IV) model development; and (V) model validation. The risk models were generated as previously described [4]. Briefly, after feature normalisation and clustering, feature selection was performed multiple times using 1000 bootstrap samples of the exploratory cohort. Subsequently, model training was conducted on 1000 bootstrap samples of the exploratory cohort, using the highest ranked features as well as the optimised hyper-parameter set. Finally, an ensemble prediction was made by averaging the predicted risk scores of each model for both the exploratory and the independent validation cohort separately.

Combinations of five feature selection methods and six learning algorithms were used for model development to reduce the risk of incidental findings, based on the recommendation in Leger et al. [4]. The following feature selection methods were used: Spearman correlation, mutual information maximisation (MIM), mutual information feature selection (MIFS), minimum redundancy maximum relevance (MRMR) and random forest variable importance (RFVI). The six learning algorithms comprised: Cox proportional hazard model (Cox), boosted tree-Cox (BT-Cox), boosted gradient linear model-Cox (BGLM-Cox), random survival forest (RSF) and maximally selected rank statistics random forest (MSR-RF) as well as the full-parametric BT-Weibull model. Table A3 summarises the definition of the hyper-parameters of the feature selection methods and of the machine learning algorithms, which were used during the hyper-parameter optimisation.

### 2.4. Performance Assessments

(I) The prognostic performance of the radiomic models was assessed on the exploratory and on the independent validation cohort using the concordance index (C-index) [26,27]. The C-index is a generalisation of the area under the curve for continuous time-to-event survival data and a C-index of 0.5 describes a random prediction, whereas a perfectly predicting model has C-index of 1.0. Risk models were developed based on GTV${}_{\mathrm{entire}}$, GTV${}_{3mm-rim}$ and GTV${}_{5mm-rim}$, as well as the corresponding core volumes GTV${}_{3mm-core}$ and GTV${}_{5mm-core}$.

The median C-indices over all combination of feature selection methods and machine learning algorithms were determined based on the exploratory and the validation cohort for each tumour sub-volume. For the considered feature selection methods and machine learning algorithms, the model performance was statistically compared between the GTV${}_{3mm-rim}$ and GTV${}_{5mm-rim}$ and their corresponding core volumes using a multi-level model approach (MLA), which is described in Section A.1 [5]. Subsequently, representative model combinations for each sub-volume were selected, consisting of one feature section method and one learning algorithm. To choose this representative model, the median performances of every feature selection method over all learning algorithms and vice versa were determined. The model generated by the feature selection method with a C-index closest to the median feature selection performances and the learning algorithm with a C-index closest to the median learning algorithm performances was then selected. For the further analyses, the representative models based on the sub-volume of (a) the GTV${}_{\mathrm{entire}}$; (b) the tumour rim; (c) the corresponding core and (d) the extended rim were investigated in more detail. In addition, we assigned the patients of the validation cohort into two sub-groups according their initial tumour volume using 20 cm${}^{3}$ as a threshold value, which corresponds to a tumour radius of approximately 1.5 cm in the case of a spherical tumour. Subsequently, we investigated the prognostic performance of the developed models using the resulting sub-groups individually.

(II) Risk-based patient stratification into groups at low and high risk of loco-regional recurrence was performed for each tumour sub-volume and for all model combinations. The results for the selected models (a)–(d) were analysed in more detail. Patients were stratified into a low and high risk group based on the predicted risk of the radiomics models. The cut-off value used for stratification was based on the median predicted risk (median${}_{\mathrm{risk}}$) determined on the exploratory cohort. This cut-off value was directly applied to the validation cohort. Survival curves were estimated using the Kaplan–Meier method and the stratification was compared using log-rank tests. Log-rank test *p*-values < 0.05 were considered to be statistically significant.

(III) Radiomic signatures were analysed in detail for the models trained on the different selected tumour sub-volumes (a)–(d). Features included in the signatures and their expression values were depicted as heatmaps for the exploratory and the validation cohort. For this purpose, all patients were sorted according to their predicted risk and to their risk group stratification. To quantify the overall importance of the identified features, univariate significance of the individual radiomic features included in the signatures were tested by the Cox model on the entire patient cohort.

## 3. Results

The number of loco-regional recurrences was 84 for the exploratory and 28 for the independent validation cohort, respectively. The primary endpoint LRC showed no significant difference between both cohorts (*p* = 0.26). The median follow-up time was 21.2 months (range: 1.2–131.9 months) for the exploratory and 24.3 months (range: 1.3–107.2 months) for the validation cohort (*p* = 0.64). The median GTV${}_{\mathrm{entire}}$ was 29.2 cm${}^{3}$ (range: 4.4–322.2 cm${}^{3}$) in the exploratory cohort and 40.1 cm${}^{3}$ (range: 2.7–239.0 cm${}^{3}$) in the validation cohort (*p* = 0.067, Table 1). For further analyses, 3 mm and 5 mm margins were subtracted from the GTV${}_{\mathrm{entire}}$, respectively. The median volume fractions of the resulting tumour rim sub-volumes were 47.7% (range: 26.1–59.5%) for the GTV${}_{3mm-rim}$ and 52.8% (range: 33.1–59.9%) for the GTV${}_{5mm-rim}$ sub-volumes in the exploratory cohort and 46.4% (range: 26.1–59.5%) and 52.1% (range: 33.1–59.9%) in the validation cohort (*p* = 0.066 and *p* = 0.081, respectively).

### 3.1. Prognostic Performance

Radiomic models were developed and validated based on the different (sub-)volumes of the tumour. Their performance for the prognosis of LRC is summarised in Table 2 for both cohorts. The median C-index on the exploratory cohort was between 0.72 and 0.76 for the considered sub-volumes. For the validation cohort, models based on the GTV${}_{\mathrm{entire}}$ achieved a median prognostic performance of 0.61 ± 0.04 (median ± standard deviation (SD)), while the models based on the tumour rim sub-volumes showed a slightly better median performance on the validation cohort (C-index: GTV${}_{3mm-rim}$: 0.63 ± 0.03 and GTV${}_{5mm-rim}$: 0.65 ± 0.02, respectively). The core-based risk models revealed the lowest prognostic performance on the validation cohort (C-index: GTV${}_{3mm-core}$: 0.60 ± 0.02 and GTV${}_{5mm-core}$: 0.59 ± 0.01, respectively). The difference in C-index between GTV${}_{5mm-rim}$ and GTV${}_{5mm-core}$ showed a statistical trend (MLA: *p* = 0.10), while the difference between GTV${}_{3mm-rim}$ and GTV${}_{3mm-core}$ was not statistically significant (MLA: *p* = 0.50). The median performances of the sub-group analyses showed similar results for small GTV${}_{\mathrm{entire}}$ (≤20 cm${}^{3}$) between the tumour rim- and core-based models (3 mm: 0.62 vs. 0.63 and 5 mm: 0.67 vs. 0.69, respectively), whereas, the differences between rim and core were larger for larger GTV${}_{\mathrm{entire}}$ (3 mm: 0.61 vs. 0.57 and 5 mm: 0.61 vs. 0.57) in the validation cohort. Furthermore, overall performance was higher for the sub-group of smaller tumours.

The C-indices of the representative models for each tumour sub-volume are shown in Table 3. Among all GTV${}_{\mathrm{entire}}$-based risk models, the RSF algorithm in combination with the RFVI feature selection method was selected as representative model for further analysis (C-index: 0.75, 95% confidence interval [0.71–0.81]). On the validation cohort, this model achieved a C-index of 0.63 ([0.49–0.67]). The RSF–MIM model trained on the GTV${}_{5mm-rim}$ was selected as representative model compared to all other rim-based models on the exploratory cohort (C-index: 0.77, [0.72–0.82]). This model attained an improved performance on the validation cohort (C-index: 0.66, [0.52–0.69]), which was slightly higher compared to the GTV${}_{\mathrm{entire}}$-based model. The corresponding GTV${}_{5mm-core}$-based model (RSF–MIM) showed a lower prognostic performance on the validation cohort (C-index: 0.61, [0.49–0.69]) compared to the GTV${}_{5mm-rim}$ model. Figure A1 and Figure A2 show the prognostic performance for the considered feature selection methods and learning algorithms based on the GTV${}_{\mathrm{entire}}$, GTV${}_{3mm-rim}$ and GTV${}_{5mm-rim}$, as well as the corresponding core sub-volumes on the exploratory and the validation cohorts.

The GTV${}_{5mm-rim}$ sub-volume, which achieved the highest prognostic performance among all rim-based models was subsequently extended by different margins beyond the originally delineated tumour into surrounding tissue. The tumour extensions GTV${}_{5mm-rim+2mm-ext}$ and GTV${}_{5mm-rim+3mm-ext}$ showed the highest median performances on the validation cohort (C-indices: 0.63 ± 0.03 and 0.64 ± 0.05, respectively). The C-index of the representative models trained on the different tumour extensions are shown in Table 2. The representative model trained on the GTV${}_{5mm-rim+3mm-ext}$ (RSF–RFVI) achieved a slightly better performance in validation compared to the model based on the GTV${}_{\mathrm{entire}}$ and on the GTV${}_{5mm-rim}$ (C-index: 0.67, [0.60–0.77]). The resulting C-indices for all extended sub-volumes and developed radiomic risk models on the exploratory and validation cohort are summarised in Figure A3 and Figure A4, respectively. The hyper-parameters including the optimised values for the representative models are given in Table A4.

### 3.2. Risk-Based Patient Stratification

Patients were stratified into groups at low and high risk for loco-regional recurrence based on the model prediction of the exploratory cohort. Table 3 shows the *p*-values of the log-rank test for LRC for all representative models on the validation cohort based on the median${}_{\mathrm{risk}}$ cut-off. The RSF–RFVI model trained on GTV${}_{\mathrm{entire}}$ was able to stratify patients into low and high risk groups with a significant difference in LRC (*p* = 0.012). A slightly improved stratification could be achieved by the GTV${}_{3mm-rim}$- and GTV${}_{5mm-rim}$-based models (*p* = 0.005 and *p* = 0.006, respectively) as well as by the extended volume GTV${}_{5mm-rim+3mm-ext}$ (*p* < 0.001). Stratification based on the predicted risk of the corresponding GTV${}_{5mm-core}$ model (RSF–MIM) did not lead to significant differences in LRC between both groups (*p* = 0.11).

Figure 2 shows the Kaplan–Meier curves using the median${}_{\mathrm{risk}}$ cut-off for the representative models based on (a) GTV${}_{\mathrm{entire}}$, (b) GTV${}_{5mm-rim}$, (c) GTV${}_{5mm-core}$ and (d) GTV${}_{5mm-rim+3mm-ext}$, respectively, for the validation cohort. The resulting *p*-values for all considered sub-volumes and developed radiomic risk models on the validation cohort are summarised in Figure A5 and Figure A6.

### 3.3. Radiomic Signature Analysis

Radiomics signatures were investigated for the representative models based on GTV${}_{\mathrm{entire}}$, GTV${}_{5mm-rim}$, GTV${}_{5mm-core}$ and GTV${}_{5mm-rim+3mm-ext}$. Figure 3 shows the feature expressions of the developed signatures for each patient in a heatmap. Image features within the signatures are listed in Table A5.

The developed signatures for the different models (a)–(d) consist of two to ten imaging features extracted from the original and wavelet transformed images. The selected features typically comprise first-order statistical or texture-based features. For instance, the ’statistics energy’ feature, which describes the overall density of the tumour volume, appears in all four signatures as a single or as a feature within a cluster [2]. Furthermore, the signatures of the trained models (a) and (b) consist of the same intensity-volume histogram feature (i.e., ‘ivh_diff_v10_v90’) computed and extracted from the wavelet transformed images. This feature describes the difference between the largest volume fractions at two different intensity values of at least 10% and 90% [25,28]. The selected features for (a) and (b) were mostly based on the low-pass wavelet transformed images, which may contain reduced noise. Features within the signatures (c) and (d) were mostly computed on the high-pass wavelet transformed images, which may characterise edges and blobs within the considered regions. For all developed signatures (a)–(d), almost all features were significantly associated with LRC based on univariate Cox analyses using the entire patient cohort.

## 4. Discussion

Tumours may contain biologically complex structures and exhibit substantial spatial variation. Thus, the main objective of this study was to compare radiomic models based on different sub-volumes of the tumour, i.e., on the tumour rim, the tumour core and the macroscopic tumour extensions, in order to identify potential regions containing the most relevant prognostic information for LRC. Using CT imaging of patients with locally advanced HNSCC revealed that radiomic risk models based on tumour rim sub-volumes achieved a slightly improved prognostic performance and better patient stratification compared to models based on the corresponding core regions. Furthermore, sub-group analyses showed that the differences in prognostic performance between rim and core regions were larger for large tumours compared to small tumours. In general, our analysis showed a good median performance and a better patient stratification for the models based on the GTV${}_{5mm-rim}$, while the corresponding core-based models performed slightly less. This may indicate that the tumour rim contains more prognostic information. The statistical comparison between both sub volumes led to a borderline statistical trend (MLA: *p* = 0.10), i.e., the presented findings require additional validation in the future.

These results are in-line with previously published data of other tumour entities [10,11,29]. For example, Dou et al. [30] showed that models based on CT-imaging features of the 3 mm rim of the GTV lead to an improved prediction of distant metastasis compared to the model based on the entire GTV for patients with locally advanced non-small cell lung cancer (NSCLC). Furthermore, Hosney et al. [31] developed a deep learning-based prediction model using a 3D convolutional neuronal network for the prediction of OS for NSCLC patients and observed that the network tended to focus on the interface between the tumour and stroma (parenchyma or pleura) regions in the CT images. In contrast to that, Keek et al. [32] concluded that the consideration of the tumour rim did not lead to an improved prediction of overall survival, loco-regional recurrence and distant metastases in stage III and IV HNSCC patients. However, for the prediction of loco-regional recurrence, a better prediction could be observed for the 5 mm rim-based model compared to the model using the GTV${}_{\mathrm{entire}}$ in the exploratory and validation cohort (C-index: 0.86/0.59 and 0.81/0.52, respectively). In addition, Grove et al. [18] showed that tumour-rim-based radiomic features (i.e., entropy) were higher expressed compared to features extracted from corresponding tumour-core sub-volumes in NSCLC patients. While, the entropy feature and their ratios of core and rim regions were associated with overall survival in the exploratory cohort, but not in the independent validation cohort.

The biological characteristics of the tumour rim were already discussed by published data from the Danish Head and Neck Cancer (DAHANCA) group [33]. Based on experience from pathological examination of surgical resections, the DAHANCA group concluded that for primary tumours, the risk of sub-clinical microscopic spread was around 50% of which more than 99% was within 5 mm and 95% within 4 mm of the rim of the primary tumour. Furthermore, Apolle et al. [22] showed that most solid tumours exhibit microscopic tumour extension in particular for head and neck cancer. Our findings suggest that biological processes such as microscopic spread capacity are associated with macroscopic CT imaging.

Defining the precise extent of the macroscopic tumour prior to and during RCT is difficult, especially using CT imaging without contrast enhancement [34]. Slight extensions of the delineated tumour volume into normal tissue did not reduce the performance of the radiomic risk models, which indicates that these regions may also contain prognostic information. In addition, slight extensions of the tumour may be useful for assessing feature stability, simulating different tumour delineations of different observers [35]. Furthermore, uncertainties in the delineation of the GTV${}_{\mathrm{entire}}$ may affect radiomic features and in turn the results of radiomic analyses. In the current study, the GTV${}_{\mathrm{entire}}$ was manually delineated by one expert radiation oncologist. The consideration of multiple tumour delineations of different experts or the usage of semi-automatic segmentation algorithms as well as contour randomisation techniques may help to increase the robustness of the radiomics features and improve the corresponding risk model performance, which should be investigated in the future [35,36,37].

The presented study is motivated form the assumption that hypoxic or necrotic regions preferantly appear in the tumour core due to inadequate vascular supply, and that proliferating cancer cells mainly occur in the tumour periphery [38]. Our retrospective patient cohort contains tumours with a wide range of different volumes. While necrotic and hypoxic regions will be minimal in small tumours they may be substantial in larger tumours, i.e., the prognostic value of tumour core and rim may change depending on the tumour volume. Therefore, we performed a subgroup analysis considering patients with small and large tumours separately. We found larger differences in prognostic performance between rim and core regions for larger tumours in validation, supporting this hypothesis. Still, the inclusion of patients with small and large tumours in our main analyses may affect the difference in performance between the rim- and core-based risk models. Moreover, necrotic/hypoxic regions may be heterogeneously distributed in the tumour and not be sufficiently captured by our simple approach of defining the tumour core and rim, which in addition does not consider other complex spatial and temporal variations in the tumour micro-environment. This may lead to smaller observable differences in the performance between the rim- and core-based models [14,39]. The identification and incorporation of tumour specific regional variations by more sophisticated image analysis techniques may help to overcome this gap. For instance, differential information from multi-modal imaging data, such as PET-CT or functional MRI may be used. Moreover, super-voxel algorithms can be applied to group voxels into super-voxel segments based on their grey value, e.g., using the FDG uptake value [40]. Subsequently, the resulting super-voxel segments can be further merged to generate tumour sub-volumes, e.g., by hierarchical or fuzzy c-means clustering algorithms across the entire patient cohort. Wu et al. [19] proposed such a two-stage clustering process, for the identification and determination of sub-volumes based on CT imaging combined with FDG-PET scans in lung cancer patients. Furthermore, the consideration of regions with temporal changes, e.g., due to RCT-induced tumour shrinkage or re-oxygenation using in-treatment CT images in combination with functional imaging may offer the potential to enhance radiomic risk models in future [5,39]. However, due to missing functional imaging, it was not possible to use such imaging data in this study.

In addition to radiomic features, clinical parameters may be relevant for the prediction of treatment outcome. On our cohort, from the parameters shown in Table 1, only the primary tumour volume and the derived tumour sub-volumes were significantly related to LRC (*p* < 0.01). These parameters revealed C-indices between 0.62 and 0.63 in the validation cohort using univariable Cox regression model. This was slightly lower than observed for the presented radiomic models based on the GTV${}_{5mm-rim}$ and the GTV${}_{5mm-rim+3mm-ext}$ (C-index: 0.65 and 0.67, respectively). While the radiomic signature based on the GTV${}_{5mm-rim}$ contained two CT features with a strong Spearman correlation (ρ) to the tumour volume (ρ > 0.85), the features of the signature based on the GTV${}_{5mm-rim+3mm-ext}$ showed only moderate correlations to the tumour volume (ρ range: [−0.61–0.40]). This indicates that additional imaging features, which are not related to the tumour volume, may improve the risk model performance.

One limitation of this retrospective study is the different distribution of the clinical characteristics between the exploratory and validation cohort, e.g., in tumour site and UICC stage (Table 1). Despite these differences, the validation of the presented radiomic models was successful, and due to the definition of both cohorts that was based on independent clinical trials, the presented results should be more robust compared for example to a random split of the data. Furthermore, other factors related to the retrospective nature of our study may have implications on the presented results [41]. For instance, the variety of different CT imaging acquisition and reconstruction parameters may affect the feature robustness and thereby the results of risk modelling (Table A1). Therefore, open and standardised protocols for image acquisition, reconstruction, and analysis may help to increase the robustness of radiomic risk model [42,43,44]. In addition, the biological meaning of the selected imaging features within the developed signatures and the differences between the features of the tumour rim and core remains still unclear. Therefore, these open questions should be investigated systematically in the future for a better understanding of the underlying mechanisms.

## 5. Conclusions

In the present study, we showed that radiomic models based on the rim of locally advanced HNSCC achieved a slightly higher prognostic performance for LRC after primary radio-chemotherapy compared to models using the tumour core. This supports our initial hypothesis that the tumour rim is biologically more diverse and important treatment-related processes occur primarily in this region, which may be visible in clinical imaging data. Therefore, after additional prospective validation the consideration of tumour sub-volumes may be a promising way to improve prognostic radiomic risk models.

## Figures and Tables

**Figure 1 cancers-12-03047-f001:**
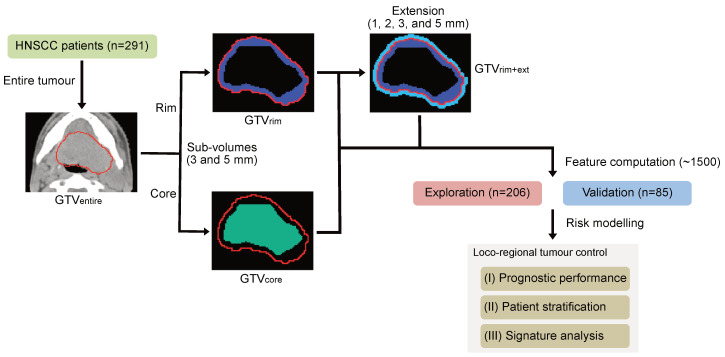
Experimental design. A multi-centre cohort of 291 patients with loco regionally advanced head and neck squamous cell carcinoma (HNSCC) was used to generate different sub-volumes based on the delineated entire tumour. In particular, the outer contour of the entire primary cross tumour volume (GTV${}_{\mathrm{entire}}$) was cropped by different margins (3 and 5 mm) to define the rim of the tumour (GTV${}_{\mathrm{rim}}$) and the corresponding core (GTV${}_{\mathrm{core}}$). Furthermore, the best performing tumour rim sub-volume was extended (GTV${}_{rim+ext}$) into surrounding tissue with different distances (1, 2, 3, and 5 mm) to assess the prognostic performance of the microscopic tumour extension. The entire cohort was split into an exploratory and an independent validation cohort for risk modelling. Prognostic model performance and patient risk group stratification were assessed on the validation cohort. Selected features within the developed signatures were analysed in terms of their univariate association with loco-regional tumour control using the entire cohort.

**Figure 2 cancers-12-03047-f002:**
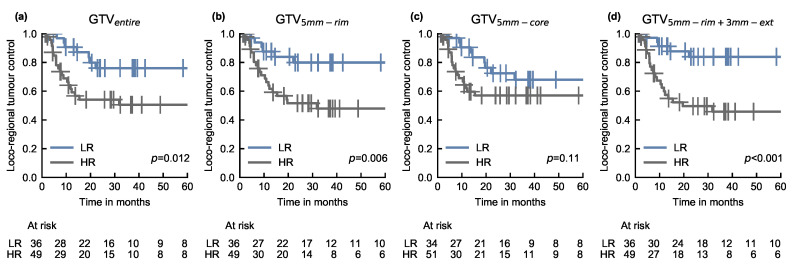
Kaplan–Meier curves for the prediction of loco-regional tumour control of the representative models based on the (**a**) entire primary gross tumour volume (GTV${}_{\mathrm{entire}}$); (**b**) 5 mm rim of the tumour (GTV${}_{5mm-rim}$); (**c**) corresponding tumour core (GTV${}_{5mm-core}$) and (**d**) 3 mm extension of the 5 mm tumour rim (GTV${}_{5mm-rim+3mm-ext}$) sub-volumes for patients of the validation cohort. Patients were stratified into low (LR) and high (HR) risk groups based on the median risk of loco-regional recurrence determined on the exploratory cohort.

**Figure 3 cancers-12-03047-f003:**
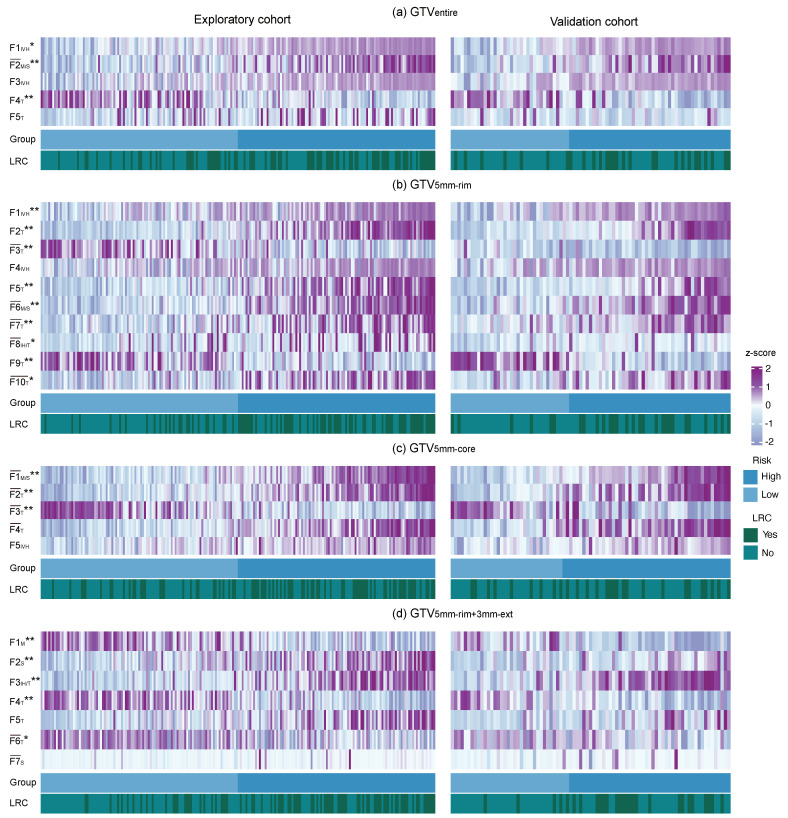
Heatmaps showing different expression patterns of the radiomic features of the developed signatures for the representative models based on the (**a**) entire primary gross tumour volume (GTV${}_{\mathrm{entire}}$); (**b**) 5 mm rim of the tumour (GTV${}_{5mm-rim}$); (**c**) corresponding tumour core (GTV${}_{5mm-core}$); and (**d**) 3 mm extension of the 5 mm tumour rim (GTV${}_{5mm-rim+3mm-ext}$) sub-volumes. Feature expression values are sorted according to the predicted risk and the risk group based on the determined median${}_{\mathrm{risk}}$ cut-off values. Loco-regional tumour control (LRC) during follow-up (yes, light; no, dark) and features with a significant association with LRC are shown (* *p* < 0.05 and ** *p* < 0.001). A detailed description of the feature abbreviations can be found in Table A5. Abbreviations: $\bar{F}$ cluster feature consisting of several features represented by the mean value as a new meta-feature, $F_{S}$ first order statistical feature, $F_{M}$ morphological, $F_{\mathrm{IH}}$ intensity histogram, $F_{\mathrm{IVH}}$ intensity volume histogram and $F_{T}$ texture feature.

**Table 1 cancers-12-03047-t001:** Patient characteristics of the exploratory and the independent validation cohort.

Clinical Variable	Exploratory Cohort	Validation Cohort	*p*-Value
Number of patients	206	85	-
Gender			
male	174	74	0.70 ${}^{2}$
female	32	11	
Age in years
median	59.0	55.0	0.023 ${}^{3}$
range	39.2–84.5	37.0–76.0	-
cTN staging
T stage 1/2/3/4	2/23/51/130	2/9/30/44	0.21 ${}^{1}$
N stage 0/1/2/3/missing	30/7/154/15/0	9/8/64/3/1	0.097 ${}^{1}$
UICC stage 2010
I/II/III/IV	0/0/15/191	1/2/9/73	0.039 ${}^{1}$
GTV (cm${}^{3}$)
median	29.1	40.6	0.067 ${}^{3}$
range	4.5–321.7	2.7–239.1	-
Tumour site
oropharynx/oral cavity/
hypopharynx/larynx	93/51/62/0	29/23/28/5	0.003 ${}^{3}$
p16 status
negative/positive/missing	148/28/30	52/5/28	0.26 ${}^{1}$
Loco-regional tumour recurrence	84 (41%)	28 (33%)	0.26 ${}^{3}$
Follow up time (months)
median	21.2	24.3	-
range	1.2–131.9	1.3–107.2	0.64 ${}^{3}$

Abbreviations: T, clinical tumour stage; N, clinical nodal stage; UICC, Union internationale contre le cancer; Gy, Gray; DNA, deoxyribonucleic acid; GTV, primary gross tumour volume; ^1^
$\chi^{2}$ test; ^2^ exact Fisher test; ^3^ Wilcoxon–Mann–Whitney test.

**Table 2 cancers-12-03047-t002:** Median concordance indices (C-index) of radiomic models using the entire gross tumour volume (GTV${}_{\mathrm{entire}}$) and the different sub-volumes, i.e., tumour rim (GTV${}_{\mathrm{rim}}$), extended tumour rim (GTV${}_{rim+ext}$) and tumour core (GTV${}_{\mathrm{core}}$) for the endpoint loco-regional tumour control. Results are presented for the exploratory and the validation cohort. Median results over all feature selection methods and learning algorithms are shown (top) as well as C-indices of the representative model combinations and the *p*-values of the log-rank tests of stratified patient groups (bottom).

Tumour Sub-Volume	Exploratory Cohort			Validation Cohort		
		All		All	GTV≤20 cm${}^{3}$	GTV>20 cm${}^{3}$
		(*n* = 206)		(*n* = 85)	(*n* = 20)	(*n* = 65)
GTV${}_{\mathrm{entire}}$		0.75 ± 0.05		0.61 ± 0.04	0.61 ± 0.07	0.59 ± 0.02
GTV${}_{3mm-rim}$		0.76 ± 0.06		0.63 ± 0.03	0.62 ± 1.00	0.61 ± 0.02
GTV${}_{3mm-core}$		0.74 ± 0.06		0.60 ± 0.02	0.63 ± 0.05	0.57 ± 0.02
GTV${}_{5mm-rim}$		0.76 ± 0.06		0.65 ± 0.02	0.67 ± 0.07	0.61 ± 0.01
GTV${}_{5mm-core}$		0.72 ± 0.04		0.59 ± 0.01	0.69 ± 0.07	0.57 ± 0.04
GTV${}_{5mm-rim+1mm-ext}$		0.76 ± 0.06		0.62 ± 0.03	0.58 ± 0.07	0.65 ± 0.03
GTV${}_{5mm-rim+2mm-ext}$		0.76 ± 0.07		0.63 ± 0.03	0.67 ± 0.04	0.66 ± 0.04
GTV${}_{5mm-rim+3mm-ext}$		0.75 ± 0.08		0.64 ± 0.05	0.61 ± 0.05	0.65 ± 0.05
GTV${}_{5mm-rim+5mm-ext}$		0.75 ± 0.07		0.63 ± 0.04	0.65 ± 0.06	0.62 ± 0.05

GTV: primary gross tumour volume; sd: standard deviation; CI: confidence interval.

**Table 3 cancers-12-03047-t003:** Concordance indices (C-index) of the representative radiomic combinations and the *p*-values of the log-rank tests of stratified patient based on the entire gross tumour volume (GTV${}_{\mathrm{entire}}$) and the different sub-volumes, i.e., tumour rim (GTV${}_{\mathrm{rim}}$), extended tumour rim (GTV${}_{rim+ext}$) and tumour core (GTV${}_{\mathrm{core}}$) for the endpoint loco-regional tumour control. Results are presented for the exploratory and the validation cohort.

Representative Model	Exploratory Cohort			Validation Cohort		
		(*n* = 206)			(*n* = 85)
	C-Index	95% CI	*p*-Value	C-Index	95% CI	*p*-Value
GTV${}_{\mathrm{entire}}$						
RSF-RFVI	0.75	[0.71–0.81]	<0.001	0.63	[0.49–0.67]	0.012
GTV${}_{3mm-rim}$						
BT-Cox-MRMR	0.76	[0.71–0.82]	<0.001	0.63	[0.52–0.70]	0.005
GTV${}_{3mm-core}$						
BT-Cox-MIM	0.75	[0.70–0.80]	<0.001	0.63	[0.50–0.70]	0.069
GTV${}_{5mm-rim}$						
RSF-MIM	0.77	[0.72–0.82]	<0.001	0.66	[0.52–0.69]	0.006
GTV${}_{5mm-core}$						
RSF-MIM	0.71	[0.66–0.77]	<0.001	0.61	[0.49–0.69]	0.11
GTV${}_{5mm-rim+3mm-ext}$						
RSF-RFVI	0.77	[0.69–0.80]	<0.001	0.67	[0.60–0.77]	<0.001

GTV: primary gross tumour volume; sd: standard deviation; CI: confidence interval; RSF: random survival forest; BT-Cox: boosted tree-Cox proportional hazard model; RFVI: random forest variable importance; MRMR: minimum redundancy maximum relevance; MIM: mutual information maximisation.

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
