# Peer review of "Comprehensive Analysis of Tumour Sub-Volumes for Radiomic Risk Modelling in Locally Advanced HNSCC"

_cancers, 2020, doi:10.3390/cancers12103047_

Round 1
Reviewer 1 Report
Excellent paper with a novel concept. Data here provides groundwork for further interesting work by this group and others.
Author Response
We thank the reviewer for this positive comment.
Please see the attachment.

Reviewer 2 Report
I would like to congratulate the authors for their outstanding research work. The article is novel, very well-written, and uses adequate methodology to answer the study questions. I just have some minor comments:
- Include the sample size, # of centers, design (retrospective), and study years in the abstract.
- Consider including a sub-section or paragraph in the methodology section that explains how the outcome (loco-regional recurrence) was ascertained - definition, methods, etc.
- Table 2 in the results section is too large, and difficult to follow. Consider splitting the table in two, one for tumor sub-volumes, and one for the representative model.
- Although the C-index for radiomic models based on rim appears higher to other models, the 95% CIs are overlapped for all the models. Therefore, it is possible that differences across models may be explained by chance alone. Can you clarify throughout the abstract and results about this? Can you comment in the discussion about this aspect?
- There is not comment in the discussion in how the retrospective design affects the study results. Also, to what population can the study results be extrapolated? Can you comment about external validity?
Reviewer 3 Report
This study investigated the effect of different tumor sub-volumes on radiomic model performance for risk prediction of locally advanced HNSCC. The radiotherapy GTVentire was cropped by different margins to define the rim and the corresponding core sub-volumes of the tumor, yielding 4 different subvolumes for radiomics investigation. The best subvolume was further extended with 4 different margins to investigate “microenvironment”. As a secondary endpoint, the effects of different feature selection methods and learning algorithms were also reported in the appendices.
The investigation is meaningful and the multi-center cohort is also nice to have. The manuscript is well written. I have the following comments.
- Though some differences in achieved CIs were observed for different subvolumes, the magnitude was very small. Since no statistical comparison was conducted between different subvolumes, how can you conclude that rim is more prognostic than the core? In other words, in such a complex study, can the small difference come from noise or random fluctuation instead?
- In Table 2, where are the results for subvolumes with extension on the validation cohort?
- For creating the GTV, was there any other imaging modality used in tandem? i.e. Did any patient have PET fusion etc for GTV contouring?
- What are the implications of segmentation uncertainty on your study results?
- Can you also report the tumor size information for your study cohort?
- Patient heterogeneity is still expected to be large among your cohorts in terms of interpatient subvolume size/shape etc, and the degree of necrosis within the tumor. Can you comment on the implications of such heterogeneity on your study results?
- Some information was reported in Table A4 for selected features. But I am not sure if any of these are shape features (e.g. is moreph_integ_int a shape feature?). Since the rim vs. core subvolume generation is based on uniform cropping/extension instead of intensity-based thresholding or other more advanced methods, what are the implications?
